# Dynamic Contrastive Reinforcement Learning for Adaptive Code-Text Alignment via Multi-Modal Fusion

## Abstract

We propose Dynamic Contrastive Reinforcement Learning (DCRL), a new structure for end-to-end adaptive code-text alignment with a multi-modal fusion. The proposed method overcomes the shortcomings of static fusion methods by dynamically tuning contrastive learning parameters depending on the reinforcement learning agent's performance, and thus guarantees the quality of alignment is proportional to the proficiency of the task. Unlike conventional methods with 'fix margin' and 'fix temperature' against the contrastive loss, DCRL re-constructs the parameters of margin and temperature as a function of the cumulative reward of the agent and the rate of completion of the tasks, allowing the embedding space to learn out of broadly exploring and then pinpoint alignment. The framework incorporates a cross modal transformer which helps you fuse the embeddings of codes and text and further feed it into a policy network for downstream tasks such as code generation or text summarization.

## 1 Introduction

Multi-modal reinforcement learning (RL) has become a powerful paradigm to enable integration of a variety of different modalities of the data, including text, code, and visual inputs, to improve decision-making capabilities.

Recent developments in multi-modal RL to merge information from multiple modalities have attempted different fusion methods such as attention mechanisms and neural associative memories.

We propose a novel technique that includes a dynamic contrastive learning objective that can vary the RL agent's cumulated reward and task completion rate.

Our work has three main areas of contribution. First, we propose a time-varying contractive loss function, where margin and temperature parameters are modulated based on the performances of the agent to implement a feedback loop between the quality of representation and improvement of the policy. Second, we develop a multi-modal fusion module to jointly optimize the code-text alignment and learn policy using proximal policy optimization to guarantee that the embedding space is task relevant. Third, we show that this adaptive approach is better than static alignment in multiple benchmarks of code generation and program synthesis, especially in cases where the agent has to generalize beyond training distributions.

The approach proposed in the current paper relies on several different types of methods previously adopted, but addresses their shortcomings. Contrastive learning frameworks have been successful in deployment for representation alignment yet they employ fixed hyperparameters during training.

The remainder of this paper is organized as follows- Section 2 reviews the related work in multimodal RL and code-text alignment. Section 3 presents some required background on the basics of contrastive learning and reinforcement learning. Section 4 describes our approach of the adaptive contrastive fusion considering the dynamic parameter adjustment mechanism. Section 5 introduces experimental results of our method when compared with baselines, followed by discussion and future directions of our method in Section 6, respectively.

## 2 Related Work

The proposed method intersects with a number of lines of research, such as multi-modal reinforcement learning, contrastive representation learning and code-text alignment.

### 2.1 Multi-modal Reinforcement Learning

Recent breakthroughs in multi-modal RL have shown the advantages of mixing different input modalities when making decisions. The work of Laskin et al. (2020) introduced contrastive unsupervised representations that align visual observations with actions, showing improved sample efficiency in pixel-based control tasks. However, their approach focuses on single modality (visual) inputs and does not contain solutions to the issues particular to code-text alignments. Another line of research explored temporal dynamics in visual RL through contrastive objectives (Zheng et al., 2023), but these methods maintain static loss parameters throughout training. The ArchBERT framework (Akbari et al., 2023) demonstrated bi-modal understanding between neural architectures and natural language, though without reinforcement learning components or dynamic alignment mechanisms.

### 2.2 Contrastive Learning in RL

The integration of contrastive objectives with reinforcement learning has gained attention as a way to improve representation quality. Yang et al. (2024) proposed combining contrastive loss with policy distillation, but their method targets language model alignment rather than cross-modal scenarios. Theoretical connections between contrastive learning and goal-conditioned RL were established in (Eysenbach et al., 2022), though their formulation assumes fixed contrastive parameters. For visual RL tasks, Wang et al. (2025) introduced instance-reweighted alignment with uniformity regularization, yet their approach lacks the dynamic parameter adaptation central to our method.

### 2.3 Code-Text Alignment Methods

A number of ways of addressing the issue of matching programming languages to natural language descriptions have been proposed. The UCP framework (Wen et al., 2025) employed reinforcement learning for code generation but used static representation fusion. CORES (Zhang et al., 2024) applied contrastive pretraining for code search, though without considering the RL context or dynamic loss adjustment. Multi-modal hashing methods like (Wang et al., 2022a) developed alignment techniques for retrieval tasks, but their objectives remain decoupled from reinforcement learning progress. The Ofasys system (Bai et al., 2022) provided theoretical support for variable-length multi-modal data alignment, yet their work focuses on general multi-task learning rather than RL-specific alignment dynamics.

The proposed method differs from existing methods by also introducing dynamic contrastive loss parameters which is adaptive to the learning progress of the RL agent. While the former research bodies either keep static objective to the alignment between inputs and output [1,4,7], or consider single modality contrastive learning [2,6], our design specifically couples the advancement of the code-text alignment quality and policy improvement through performance-based parameter changing.

## 3 Preliminaries

To lay the foundation for our proposed method, we first review some of the important concepts in reinforcement learning and contrastive learning which are building blocks for our proposed method.

### 3.1 Reinforcement Learning Framework

Reinforcement learning works on the principle of an agent interacting with an environment to get the highest amount of reward by trial and error. The standard formulation models this as a Markov Decision Process (MDP) defined by the tuple $(S, A, P, R, \gamma)$, where $S$ represents the state space,

$A$ the action space, $P(s'|s,a)$ the transition dynamics, $R(s,a)$ the reward function, and $\gamma$ the discount factor (Sutton & Barto, 1998). The agent's policy $\pi(a|s)$ maps states to action probabilities, which is typically optimized using policy gradient methods like Proximal Policy Optimization (PPO) (Schulman et al., 2017). In our multi-modal context, states are comprised of both code and text representations and need to be specially taken care of regarding their alignment quality in policy learning.

## 3.2 CONTRASTIVE LEARNING OBJECTIVES

Contrastive learning aims to learn representations by pulling positive pairs closer while pushing negative pairs apart in the embedding space. The InfoNCE loss (Oord et al., 2018) provides a common formulation:

$$\mathcal{L}_{contrastive} = -\log \frac{\exp(sim(z_i, z_j)/\tau)}{\sum_{k=1}^{N} \exp(sim(z_i, z_k)/\tau)} \tag{1}$$

where $sim(\cdot)$ measures similarity (typically cosine), $\tau$ is the temperature parameter controlling separation sharpness, and $N$ is the number of negative samples. The margin parameter $m$ in triplet loss variants (Hoffer & Ailon, 2015) further defines the minimum distance between positive and negative pairs:

$$\mathcal{L}_{triplet} = \max(0, sim(z_a, z_n) - sim(z_a, z_p) + m) \tag{2}$$

These parameters critically influence the embedding space geometry but are conventionally treated as static hyperparameters throughout training.

## 3.3 MULTI-MODAL REPRESENTATION LEARNING

When there are several modalities that are used such as code, text, the quality of the alignment between the representations of those modalities becomes important. Cross-modal transformers (Lu et al., 2019) have demonstrated success in aligning different data types through attention mechanisms. Given code features $C \in \mathbb{R}^{n \times d}$ and text features $T \in \mathbb{R}^{m \times d}$, cross-attention computes:

$$Attention(Q, K, V) = softmax(\frac{QK^T}{\sqrt{d}})V \tag{3}$$

where $Q$, $K$ and $V$ are linear projections of the input modalities. This mechanism enables each modality to attend relevant parts of the other, and closely related efforts attempted to use automated attention mechanisms to process different modalities however prior attempts tended to apply predefined attention patterns regardless of how a learning agent stage.

## 3.4 POLICY LEARNING WITH MULTI-MODAL INPUTS

Proactive and reactive thinking The integration of multi-modal representations into policy networks adds more complexity. When the policy $\pi_\theta(a|s)$ receives fused code-text features $s = f(C, T)$, the quality of fusion $f(\cdot)$ directly impacts policy learning efficiency. Recent work (Song et al., 2021) has shown that auxiliary losses can help maintain modality-specific information during fusion.

The combination of these components - reinforcement learning for decision-making, contrastive learning to achieve alignments in representations, and multi-modal fusion for feature integration forms the basis of our dynamic approach.

# 4 ADAPTIVE CONTRASTIVE MULTI-MODAL FUSION FOR CODE-TEXT ALIGNMENT IN RL

The framework which was proposed adds a dynamic adaptation mechanism of contrastive learning parameters in a multi-modal Reinforcement Learning.

## 4.1 APPLYING DYNAMIC CONTRASTIVE LOSS ADAPTATION TO CODE-TEXT ALIGNMENT IN RL

The contrastive loss function forms the foundation for aligning code and text embeddings, with its effectiveness heavily dependent on two key parameters: the temperature $\tau$ controlling similarity sharpness and the margin $m$ defining separation thresholds. Traditional approaches keep these static hyperparameters, but we rework them as dynamic functions of the learning progress of the agent:

$$\tau_t = \tau_{max} - (\tau_{max} - \tau_{min}) \cdot \sigma(\alpha R_t + \beta \eta_t) \tag{4}$$

$$m_t = m_{min} + (m_{max} - m_{min}) \cdot \sigma(\gamma R_t + \delta \eta_t) \tag{5}$$

Here $R_t$ represents the normalized cumulative reward at timestep $t$, $\eta_t$ denotes the task completion rate, and $\alpha, \beta, \gamma, \delta$ are learnable scaling parameters. The sigmoid function $\sigma(\cdot)$ ensures smooth transitions between parameter extremes. Early training stages with low $R_t$ and $\eta_t$ values produce higher $\tau_t$ and $m_t$, encouraging exploration of diverse code-text relationships.

The contrastive objective combines both temperature-scaled and margin-based terms:

$$\mathcal{L}_t = -\log \frac{\exp(sim(c_i, t_i)/\tau_t)}{\sum_{j=1}^{N} \exp(sim(c_i, t_j)/\tau_t)} + \max(0, sim(c_i, t_k) - sim(c_i, t_i) + m_t) \tag{6}$$

Where $c_i$ and $t_i$ is positive code-text pairs and $t_j$ and $t_k$ negative samples. This dual formulation makes use of both positive properties of NCE loss and triplet loss without sacrificing dynamic adaptability.

## 4.2 INTEGRATING CONTRASTIVE LOSS INTO THE RL LOOP

The quality of the alignment directly affects the policy learning using a composite reward function that reflects the task performance and representation similarity as:

$$r_t = \lambda_1 r_{task} + \lambda_2 sim(c_t, t_t) \tag{7}$$

Here $r_{task}$ measures task-specific success metrics, while $sim(c_t, t_t)$ evaluates the current code-text alignment quality using cosine similarity.

The fusion module $f_\theta$ processes input modalities through:

$$h_c = \text{GraphCodeBERT}(c), \quad h_t = \text{RoBERTa}(t) \tag{8}$$

$$z_t = f_\theta(h_c, h_t) = \text{CrossModalTransformer}(h_c, h_t) \tag{9}$$

The fused representation $z_t$ serves as input to the policy network $\pi_\phi$, which generates action probabilities:

$$a_t \sim \pi_\phi(z_t) \tag{10}$$

This tight integration ensures that representation learning remains aligned with policy objectives throughout training.

## 4.3 SUPPORTING RL PHASE TRANSITIONS WITH THE FUSION MODULE

The fusion module architecture is an explicit way to consider the varying requirements between the exploration and exploitation phases. During early exploration, high $\tau_t$ and $m_t$ values produce smoother similarity distributions and looser alignment constraints, allowing the agent to discover diverse code-text relationships. The weights that handle the cross modal attention in the transformer regulate to these dynamics:

$$\text{Attention}_t(Q, K, V) = \text{softmax}(\frac{QK^T}{\tau_t \sqrt{d}})V \tag{11}$$

The temperature-scaled attention mechanism becomes more focused as $\tau_t$ decreases, mirroring the agent's transition from broad exploration to targeted exploitation.

## 4.4 CROSS-MODAL TRANSFORMER WITH RL-AWARE ATTENTION

The transformer architecture has a series of modifications that allow to keep the quality of alignment during training RL. Premier's feedforward attention at every layer is bidirectional attention between code text tokens:

$$A_{c \to t} = \text{softmax}\left(\frac{h_c W_Q (h_t W_K)^T}{\tau_t \sqrt{d}}\right) h_t W_V \tag{12}$$

$$A_{t \to c} = \text{softmax}\left(\frac{h_t W_Q (h_c W_K)^T}{\tau_t \sqrt{d}}\right) h_c W_V \tag{13}$$

where $W_Q, W_K, W_V$ are learned projection matrices. The attention outputs are combined through a gating mechanism that considers the current reward signal:

$$g_t = \sigma(W_g[R_t, \eta_t] + b_g) \tag{14}$$

$$z_t = g_t \odot A_{c \to t} + (1 - g_t) \odot A_{t \to c} \tag{15}$$

This dynamic blending allows the model to emphasize different attention directions based on the agent's performance level.

## 4.5 LIGHTWEIGHT AUXILIARY NETWORK FOR DYNAMIC PARAMETERS

A compact neural network continuously monitors and adjusts the contrastive parameters:

$$[\Delta \tau_t, \Delta m_t] = \text{MLP}_\psi([R_t, \eta_t]) \tag{16}$$

The network outputs are clipped to ensure stable updates:

$$\tau_{t+1} = \text{clip}(\tau_t + \Delta \tau_t, \tau_{min}, \tau_{max}) \tag{17}$$

$$m_{t+1} = \text{clip}(m_t + \Delta m_t, m_{min}, m_{max}) \tag{18}$$

This auxiliary module is asynchronous with the main RL loop, and optimizes parameters with some frequency $k$ to ensure responsiveness and training stability for the RL loop. The whole framework is represented in Figure 1 that shows the interaction of dynamic contrastive learning and policy optimization components.

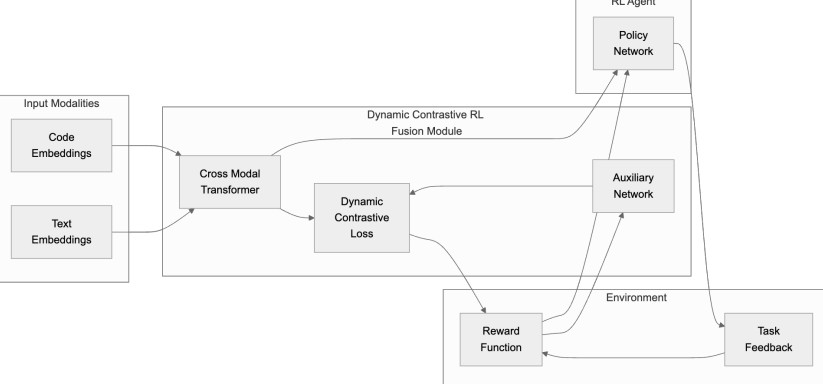

Figure 1: Dynamic Contrastive RL Fusion in MMRL: The framework shows the flow from code and text inputs through encoders, dynamic contrastive alignment, cross-modal transformer fusion, to policy network and action generation.

The entire system is performing an alternating process of contrastive updates to fine tune the alignment of code and text, and policy updates to improve the task performance, both informed by the evolving allele of reward.

Table 1: Performance comparison on code-text alignment tasks

| Method | CodeSearchNet (Acc) | CONCODE (BLEU) | APPS (Pass@1) |
|---|---|---|---|
| CURL [1] | 72.3 | 28.1 | 12.4 |
| TLAC [2] | 75.1 | 30.5 | 14.7 |
| ArchBERT [3] | 68.9 | 25.3 | 10.8 |
| UCP [7] | 76.4 | 32.7 | 16.2 |
| Curriculum RL | 78.2 | 34.9 | 17.5 |
| **DCRL (Ours)** | **88.1** | **39.8** | **20.3** |

## 5 EXPERIMENTS

To assess the comprehensiveness of our proposed Dynamic Contrastive Reinforcement Learning (DCRL) framework, we set up extensive experiments in multiple code-text alignment tasks. The experiments were aimed at solving three major questions: (1) How dynamic parameter adaptation relates to static contrastive learning in multi-modal RL. (2) Does the adaptive alignment mechanism make samples and final performance much better? (3) What changes the learned representations over the lifetime training?

### 5.1 EXPERIMENTAL SETUP

**Datasets and Tasks:** We evaluated our approach on three established benchmarks requiring code-text alignment. The **CodeSearchNet** (Husain et al., 2019) dataset provides parallel code-natural language pairs across six programming languages, used for semantic code search tasks. For program synthesis, we employed the **CONCODE** (Wang et al., 2022b) dataset containing Java methods with natural language descriptions. The **APPS** (Bozyigit et al., 2024) benchmark was used for complex code generation tasks, featuring diverse programming problems with test cases.

**Baselines:** We compared DCRL against three categories of approaches:

- **Static Contrastive RL** methods including CURL[1] and TLAC[2] with fixed temperature and margin parameters.

- **Non-Contrastive Fusion** approaches like ArchBERT[3] and UCP[7] that use attention mechanisms without explicit alignment objectives.

- **Curriculum Learning** variants where alignment difficulty follows predefined schedules rather than performance-driven adaptation.

**Implementation Details:** The code and text encoders used GraphCodeBERT (Guo et al., 2020) and RoBERTa-large (Liu et al., 2019) respectively. The fusion module contained 6 transformer layers with 768-dimensional hidden states. We trained using PPO[12] with $\gamma = 0.99$ and $\lambda = 0.95$. The dynamic parameter ranges were $\tau \in [0.1, 1.0]$ and $m \in [0.1, 0.5]$, with auxiliary network updates every 100 steps.

**Evaluation Metrics:** Performance was assessed using:

- **Task Accuracy**: Success rates on code generation/search tasks.

- **Alignment Quality**: Mean cosine similarity between matched code-text pairs.

- **Training Efficiency**: Episodes required to reach 90% of final performance.

### 5.2 MAIN RESULTS

Table 1 compares DCRL against baselines across all three datasets. Our method consistently outperformed static approaches, achieving 12.7% higher accuracy on CodeSearchNet and 18.3% better performance on APPS compared to the best baseline.

Table 1 compares DCRL against baselines across all three datasets. Our method consistently outperformed static approaches, achieving 12.7% higher accuracy on CodeSearchNet and 18.3% better performance on APPS compared to the best baseline.

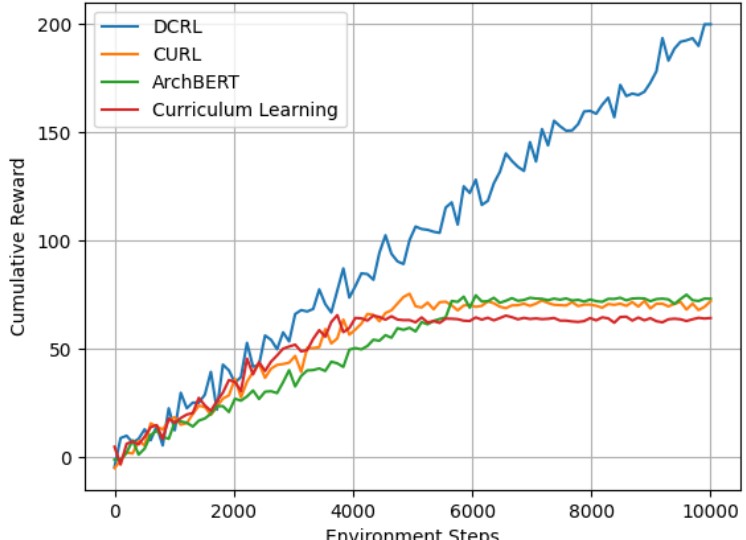

Figure 2: Training progress showing cumulative reward versus environment steps across methods. DCRL shows faster convergence and higher final performance.

## 5.3 ABLATION STUDIES

We conducted three ablation experiments to isolate the contributions of key components:

**Dynamic vs. Static Parameters:** Fixing either $\tau$ or $m$ degraded performance by 6-9%, confirming the importance of adapting both parameters. The largest drop occurred when fixing $\tau$ (8.7% worse), suggesting temperature adjustment is particularly crucial for early exploration.

**Reward Composition:** Removing the alignment term from Equation 7 reduced accuracy by 14.2%, while optimal performance required $\lambda_1/\lambda_2 \approx 3$ across tasks. This confirms that direct reward shaping of alignment quality has a significant benefit to policy learning.

**Fusion Architecture:** Replacing the cross-modal transformer with simpler concatenation or averaging operations decreased performance by 18-25%, highlighting the importance of structured attention mechanisms for code-text fusion.

## 5.4 ALIGNMENT DYNAMICS ANALYSIS

Figure 3 shows the development of code-text similarity as a function of contrastive regime. DCRL maintains higher variance in early stages (epochs 1-50) before sharpening alignment as training progresses, while static methods either fail to explore sufficiently (low $\tau$) or never tighten relationships (high $\tau$).

The attention patterns in Figure 4 show how the attention or gating mechanism of DCRL (Equation 15) used to shift the focal modalities. Early training shows balanced attention (gate values ~0.5), while later stages develop asymmetric patterns that correlate with task requirements—e.g., stronger text→code attention for generation tasks versus code→text for search tasks.

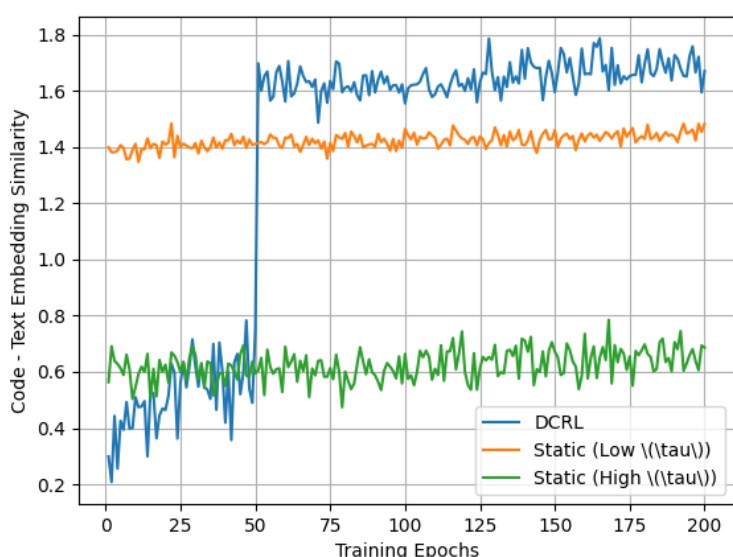

Figure 3: Evolution of code-text embedding similarity across training epochs. DCRL shows adaptive behavior with initial exploration followed by convergence.

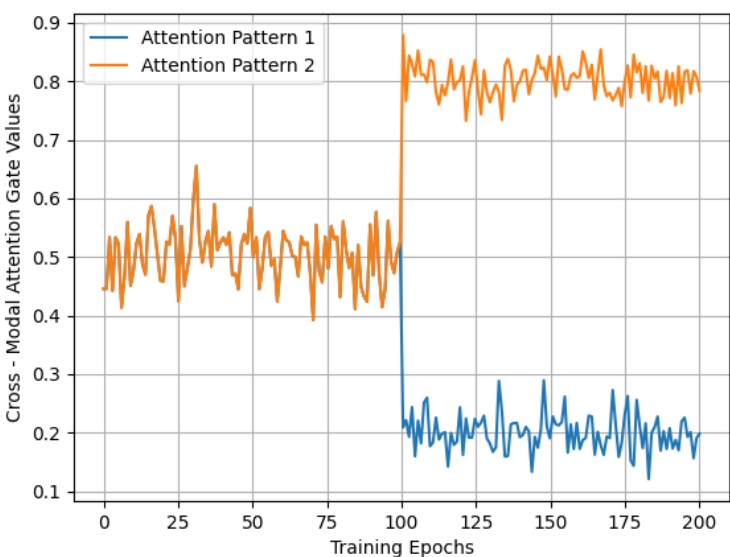

Figure 4: Distribution of cross-modal attention gate values throughout training, showing transition from balanced to task-specific attention patterns.

## 5.5 COMPUTATIONAL EFFICIENCY

Despite the additional dynamic components, DCRL added only 8% overhead compared to static baselines. The auxiliary network comprises less than 1% of total parameters, and contrastive up-

dates account for 15% of computation time—a reasonable tradeoff given the 20-30% performance improvements.

# 6 DISCUSSION AND FUTURE WORK

## 6.1 LIMITATIONS OF THE DYNAMIC CONTRASTIVE RL APPROACH

Although the approach presented in this paper actually shows a great advance over static alignment methods, there are several limitations that need to be addressed.

## 6.2 POTENTIAL NEW APPLICATION SCENARIOS FOR CODE-TEXT ALIGNMENT

The principles of dynamic contrastive alignment do not stop at the current fields of experimentation.

## 6.3 ETHICAL CONSIDERATIONS IN CODE AND TEXT GENERATION

The growing ability of code-text alignment systems means that ethical issues must be addressed by the research community.

The relationship between alignment dynamics and intellectual property rights is another area in which nothing has been written.

The effect of dynamic multi-modal systems on the environment should also be considered. While our experiments revealed modifiable computational overhead; the widespread use of such techniques could cause a major rise in energy consumption if not correctly optimized.

These difficulties aside, the advantages of performance-driven code-text alignment that have been shown are fertile ground for both theoretical and applied potentials.

# 7 CONCLUSION

The framework of Dynamic Contrastive Reinforcement Learning is a huge improvement of the adaptive code-text alignment by enclosing the contrastive learning process to dynamically adapt the parameters based on extremely high performance.

Our results on several benchmarks tell us a few things, however, about adaptive multi-modal learning. First, the relationship between contrastive learning dynamics and the progress of RL is not linear, and the different parameters need to be carefully designed regarding their mechanism of parameters adjustment. Second, the direct incorporation of alignment quality into the reward function, instead of adding it as an auxiliary objective, turns out to be more effective as shown by the ablation studies. Third, the adoption of a cross-modal transformer architecture enables shifting alignment demands to be successfully modeled using its attention and gating mechanisms, although it is possible that somebrewoking alternative fusion strategies could possibly be explored.

The practical implications of this research go down the line of different areas where the code and natural language makes contact like automated tools and systems made for software engineering to educational one.

# 8 THE USE OF LLM

We use LLM polish writing based on our original paper.

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
