# OpenReview forum: "Dynamic Contrastive Reinforcement Learning for Adaptive Code-Text Alignment via Multi-Modal Fusion"
_ICLR.cc/2026/Conference — Submitted to ICLR 2026_

### Official Review · Reviewer_gnn6 · 2025-10-15

**Soundness:** 1
**Presentation:** 1
**Contribution:** 1
**Rating:** 0
**Confidence:** 5

**Summary:**

This paper considers a kind of RL architecture where the state is made of two modalities (specifically text and code here) aligned through a contrastive loss and combined through a gated bidirectional cross-attention mechanism. The claimed contribution lies in making the temperature and the margin a function of the return instead of considering them as fixed hyperparameters, the return being build upon a reward having an additional similarity signal, as well as scaling the cross-attention with the contrastive loss temperature (thus depending on the return) and the gating mechanism (favoring a direction between modalities depending on the return value).

**Strengths:**

The rough idea of coupling the representation learning part with the task achievement is interesting, and the claimed results of the proposed approach seem to be above the baseline.

**Weaknesses:**

### Quality of writing
There are multiple problems. Overall, the paper is not well written, which makes it quite hard to follow. Many references are “hard-coded” (number like [1] not leading anywhere, eg l.92 to 96 or l.299 to 306), a paragraph is written twice (l.322 to 326), there are sentences being seemingly unfinished (eg l.440, “there are several limitations that need to be addressed”, which call for discussing these limitations, not for a new paragraph), etc.

### Clarity of writing
It is very hard to tell what the paper is about, even after reading it. The paper never defines core concepts such as what is code-text alignement, which has to be guessed from the overall architecture (unclear) presentation. It’s about reinforcement learning, but the paper never (really never) states what are the actions, what is the dynamics, what tasks is considered, corresponding to what rewards, what is the policy network, what is the value network (guessing there’s one given that PPO is mentioned with the wrong citation), etc. The same holds for all considered concepts.

### Lack of motivation / insights
The proposed approach is heuristic, which is fine, but it should be motivated, insights of why it makes sense should be provided to the reader. Eg, why making the temperature of a contrastive loss a function of the (undefined, unfortunately) return would make sense, and help learning better representations or achieving higher task completion (for what task, this being never specified).

### Inconsistencies of Eqs
There are numerous problems with the Equations. They are generally not consistent (like lhs being not homogeneous to rhs), they involved undefined quantities, it is very unclear what is implemented in the end (eg between (4-5) and (16-17))

### Experiments
There is no way the experiment could be reproduce. Even the RL tasks being considered for the various dataset is never stated in the paper. The sensibility to hyper-parameters is not discussed, but their values (or their nature) is not even provided. There is no way to know what Fig 2 corresponds to (what task, how many seed, etc), it is not even mentioned in the text. Given all these problems, the claimed results can be hardly trusted.

**Questions:**

Given the state of the paper, I think it should go through a too significant revision to be reasonably considered for ICLR. However, here are a short and non-exhaustive subset of questions and remarks that should be answered in the paper, to help for a future revision.
* What is the specific RL problem being considered, with what states, actions, dynamics, rewards, etc? It seems of foremost importance to state this in an RL paper.
* Sec 4.1, $R_t$ is said to be normalized, but how? What is the “task completion rate” (what is the task btw)? Making these two parameters adaptive introduces 4 additional scaling hyperparameters, which are said to be learnable but how and according to what loss? If they are learnt, its it important to have the reward and task completion (that evolves along learning too), rather than just meta-learning the temperature? How are chosen the positive and (two kind of) negative samples for the combined contrastive loss, in the context of the undefined RL process?
* Sec 4.2, a similarity measure is added to the reward, but is this term controllable by the RL agent, why and how? It is really unclear, given that the RL problem is undefined. If it is not controllable, why adding it to the reward (just noise from the agent perspective). Terms in Eq (8) should be defined. How are defined $c$ or $t$ vs $h_c$ or $h_t$?
* Sec 4.3, Eq (11), is it the same temperature as defined before? Why so?
* Sec 4.4 (but also elsewhere) how are learned the various introduced parameters, notably those related to the gating part, according to the contrastive loss, to the RL gradient, to both, to something else?
* Sec 4.5, does this compact neural network replaces what was described in 4.1, why, how are the underlying parameters learnt, what is the corresponding architecture, etc. L. 269 what is “evolving allele of reward”?
* Sec 5.1, the part “datasets and tasks” only describes the datasets, not the tasks, which is a big problem. Are the baselines really comparable to the proposed approach, for the considered task. There does not seem to be a method from the related work “code-text alignment” (sec. 2.3, hard to tell though with broken refs) in this “code-text alignement” experiments, if so why so?
* Fig 2, what does this figure correspond to? What is represented? For example, the reward for DCRL includes a similarity component (a stated contribution of this paper), while CURL does not for example, and ArchBert is not an RL method. So is the difference in return not just a consequence of a different reward. Btw without any detail of information there is very little value in this figure.
* Fig 3 and 4, can you explain why there is such an abrupt change? Why do these changes happen at 50 or 100 epochs (seems a little bit too round)? Fig 4, can you explain why the attention patterns are exactly the same for exactly the 100 first epochs?

---

### Official Review · Reviewer_6yLH · 2025-10-28

**Soundness:** 2
**Presentation:** 2
**Contribution:** 2
**Rating:** 2
**Confidence:** 4

**Summary:**

This paper proposes a novel framework called *Dynamic Contrastive Reinforcement Learning (DCRL)* for adaptive code–text alignment. Its core innovation lies in overcoming the limitations of conventional contrastive learning methods that rely on static hyperparameters (e.g., fixed margin ( m ) and temperature ( $\tau$ )). DCRL links these parameters dynamically to the real-time performance of an RL agent—measured via cumulative reward ( R_t ) and task completion rate ( \eta_t ). This allows the model to perform broader exploration (high ( $\tau, m$ )) during early training and transition to precise alignment (low ( $\tau, m$ )) as performance improves. The framework incorporates a cross-modal Transformer to fuse code and text embeddings, optimized via PPO for downstream tasks such as code generation and program synthesis.

**Strengths:**

1. **High Originality:** Dynamically linking contrastive parameters to RL performance is a novel and insightful idea, addressing a real pain point in static methods.
2. **Clear Motivation:** The intuition behind dynamic parameters (high ( $\tau, m$ ) → exploration; low ( $\tau, m$ ) → exploitation) is well-articulated.
3. **Architectural Coherence:** Despite textual flaws, the overall DCRL architecture is logically designed—combining encoders, cross-modal Transformer, and PPO with alignment quality directly included in the reward (Eq. 7).
4. **Strong Empirical Results:** As shown in Table 1, DCRL consistently outperforms baselines across three benchmarks.
5. **Analytical Insights:** Figures 3–4 provide meaningful visualizations of the dynamic alignment process and shifting attention patterns.

**Weaknesses:**

1. **[Major] Unclear dynamic parameter update mechanism:**
    - *Mechanism A (Eqs. 4–5):* Parameters are direct functions of ( $R_t$ ) and ( $\eta_t$ ).
    - *Mechanism B (Eqs. 16–18):* An auxiliary MLP outputs parameter deltas ( $\Delta \tau_t, \Delta m_t$ ).
    - The relationship between A and B is never explained. Does B learn the coefficients ( $\alpha, \beta, \gamma, \delta$ ) in A? Or does it replace A entirely? How is the MLP trained?

        This ambiguity is the paper’s most critical flaw.

2. **[Major] Misleading framework diagram (Fig. 1):** The actual information flow and feedback loops are incorrectly depicted.
3. **Confusing use of ( $\tau_t$ ):** The same temperature variable appears in both contrastive loss (Eq. 6) and attention mechanisms (Eqs. 11–13). Are they the same? Why would an RL-driven temperature simultaneously govern attention sharpness? This dual role needs justification.

**Questions:**

1. **[Crucial] Clarify the exact update mechanism of dynamic parameters ( $\tau$ ) and ( $m$ ).**
    - Which approach (Eqs. 4–5 vs. 16–18) is actually implemented?
    - (a) If Eqs. 4–5, how are the hyperparameters ( $\alpha, \beta, \gamma, \delta$ ) set?
    - (b) If Eqs. 16–18, what is the detailed MLP architecture, its loss function, and how is its “asynchronous” update performed?
2. **Dual role of ( $\tau_t$ ):** Is it truly the same parameter in both contrastive and attention modules? Provide theoretical or empirical justification.
3. **Framework diagram:** Can the authors provide a corrected version (e.g., in the appendix) accurately showing the feedback loops for ( $R_t$ ) and ( $\eta_t$ )?

---

### Official Review · Reviewer_gxYJ · 2025-10-30

**Soundness:** 2
**Presentation:** 3
**Contribution:** 3
**Rating:** 4
**Confidence:** 4

**Summary:**

The paper introduces a method named Dynamic Contrastive Reinforcement Learning (DCRL) to address a limitation in multi-modal RL, specifically for code-text alignment. The core problem identified is that static parameters in contrastive learning (such as temperature and margin) are suboptimal as they fail to adapt to the different phases of an RL agent's training, from initial exploration to later exploitation. The proposed solution is to make these parameters dynamic, specifically by defining them as functions of the agent's performance (i.e., cumulative reward and task completion rate). This dynamic alignment mechanism is integrated within a multi-modal framework using a cross-modal transformer. Experiments conducted on three benchmarks (CodeSearchNet, CONCODE, and APPS) show that DCRL significantly outperforms baselines that use static parameters or non-contrastive fusion methods. However, The methodology provide by this paper seems not self-consistent and lack of solid experiment support. In conclusion, I think this paper needs revision and is marginally below the acceptance threshold.

**Strengths:**

The paper propose an approach in MLLM Reinforcement alignment: coupling the geometry of the representation space (controlled by contrastive parameters) directly to the policy's learning progress. This dynamic feedback loop, where the agent's success dictates the representation learning's focus, is a compelling idea. This novelty is supported by a robust experimental design, utilizing three distinct and complementary code-text benchmarks that cover search, synthesis, and complex generation tasks. Furthermore, the paper provides a thorough set of ablation studies and dynamic analyses that effectively demonstrate the importance of the dynamic parameters (especially temperature) and the inclusion of an alignment-based reward, adding significant weight to the authors' claims.

**Weaknesses:**

1) There seems to be a critical, unresolved contradiction in the methodology section. Section 4.1 and Section 4.5 present two entirely different mathematical mechanisms for updating the dynamic parameters. One (4.1) describes a direct calculation via a sigmoid function, while the other (4.5) describes an auxiliary MLP learning parameter increments. I am confusing which method brings the improvements? These two methods appear to be juxtaposed rather than logically integrated and this inconsistency makes the method seems arbitrary and fundamentally non-reproducible.

2) The experiment seems not sufficient. The introduction makes a strong claim about the method's ability to generalize "beyond training distributions." This is a significant claim, but the experimental section (Section 5) provides no out-of-distribution (OOD) experiments or evidence to substantiate it.

3) The comparative experiments in the paper appear unfair and lack further ablation studies. The performance gains reported in Table 1, particularly the large gap over the "Curriculum RL" baseline, are substantial enough to warrant scrutiny. The paper lacks explicit confirmation that all baselines were implemented under identical conditions (e.g., using the same GraphCodeBERT backbone and PPO optimizer), raising concerns about the fairness of the comparison.

**Questions:**

See weaknesses.

---

### Official Review · Reviewer_fG5a · 2025-11-03

**Soundness:** 2
**Presentation:** 2
**Contribution:** 2
**Rating:** 2
**Confidence:** 4

**Summary:**

The paper “Dynamic Contrastive Reinforcement Learning (DCRL)” introduces a method to improve how code and text are aligned in machine learning systems. Instead of keeping fixed parameters during contrastive learning, DCRL changes these parameters—such as margin and temperature—based on how well a reinforcement learning agent performs. This helps the model adjust as it learns, linking code and text more effectively during both training and downstream tasks like code generation and summarization.

The system combines contrastive learning and reinforcement learning using a cross-modal transformer that fuses code and text features. It also includes a small network that automatically updates the contrastive parameters. Experiments on datasets like CodeSearchNet, CONCODE, and APPS show that the approach performs better than older static or schedule-based methods. The authors also report that both the adaptive parameters and the transformer module are important for achieving this improvement.

In the discussion, the paper mentions some limits such as higher computation cost and the need for further analysis of how dynamic parameters affect learning. It also points out possible uses in other multi-modal tasks and briefly notes ethical concerns related to data use and energy consumption.

**Strengths:**

The paper’s main strength lies in its introduction of a dynamic contrastive learning mechanism that adjusts key parameters like margin and temperature according to the reinforcement learning agent’s progress, making the alignment process more adaptive and effective than fixed-parameter methods. It also integrates reinforcement learning and contrastive learning in a meaningful way by tying representation quality directly to the reward signal, forming a feedback loop that improves both policy and embedding learning. The experiments are covering multiple benchmarks such as CodeSearchNet, CONCODE, and APPS, with consistent improvements over strong baselines. The ablation studies clearly show the impact of each component, reinforcing the soundness of the approach. The architecture design, which combines a cross-modal transformer and an auxiliary parameter network, is thoughtfully constructed and supports stable training. Finally, the overall concept of dynamically adjusting contrastive parameters could extend to other multimodal learning tasks, giving the method potential relevance beyond code-text alignment.

**Weaknesses:**

While the paper proposes an interesting concept, several weaknesses reduce its overall clarity and scientific depth. First, the novelty of the method is somewhat overstated. The core idea—dynamically adjusting parameters like margin and temperature in contrastive learning—feels more like a tuning mechanism rather than a fundamentally new theoretical contribution. The connection between reinforcement learning performance and contrastive loss adjustment is intuitive but not deeply justified either mathematically or conceptually. The authors claim that this adaptation improves exploration and exploitation balance, but the paper does not clearly analyze why or how this behavior emerges. In essence, the framework reads more as an engineering extension of existing ideas rather than a breakthrough in algorithmic design.

Second, the experimental section, while extensive in datasets, lacks depth in analysis. The benchmarks (CodeSearchNet, CONCODE, and APPS) are standard, and results mainly show numeric improvements without discussing error types, qualitative outcomes, or where the adaptive method helps most. There is also limited comparison with recent or more advanced multimodal alignment models—especially newer LLM-based systems that already capture strong code-text alignment. Ablation studies focus on the internal components but do not explore sensitivity to parameter ranges or stability over training runs, leaving uncertainty about reproducibility and general robustness. The reported improvements, although consistent, may not justify the added complexity and computational cost of maintaining dynamic updates.

Third, the discussion of limitations and ethics feels superficial. The authors briefly mention computational cost and environmental impact but do not quantify them or propose concrete mitigation. Similarly, while they mention potential applications, there is little insight into how the approach would integrate with large-scale or real-world coding systems, where RL environments for code generation are often expensive and unstable. The framework’s dependency on continuous RL feedback could also make it difficult to scale in practice.

Finally, the writing quality significantly detracts from readability. The paper is unevenly structured, with long, cluttered sentences and unclear transitions between ideas. Many sections read like a technical report rather than a polished research paper. Key terms are introduced abruptly, and figures are poorly explained. The grammar and phrasing are inconsistent, and the tone oscillates between formal and informal. Overall, the presentation lacks precision and polish, making it harder to grasp the contributions and limiting the paper’s impact despite an interesting direction.

**Questions:**

1) How sensitive is the proposed method to the choice of scaling parameters (α, β, γ, δ) that control how margin and temperature change over time? Would different settings lead to unstable or inconsistent behavior?
2) Does the dynamic adjustment truly capture learning progress, or could it simply overfit to short-term reward fluctuations? How does the method handle noisy or sparse rewards, which are common in RL-based code generation?
3) How much of the performance improvement comes from the adaptive contrastive mechanism itself, versus from the stronger transformer backbone (GraphCodeBERT + RoBERTa)?
4) Would similar gains be observed if the same dynamic parameter adaptation were applied to other multi-modal tasks (e.g., vision-language alignment)? This would help determine if the method is generally useful or specific to code-text data.

---

### Meta-Review · Area_Chair_N3R4 · 2026-01-07

**Summary:**

All the reviewers agreed there are some major problems with this paper. The idea of changing the contrastive learning settings based on how well the RL agent is doing is interesting, but the explanation is confusing. There are two different ways described for updating the parameters and it's not clear which one they really used. Also they never properly explain the RL part like what the actions or rewards actually are. The experiments are hard to reproduce because a lot of details are missing, and the writing has some rough spots with incomplete sentences and references that don't work. So even though the core thought is good, the paper isn't really ready.

**Reviewer Concerns:**

Since there wasn't any author reply, none of the worries got answered. The main issues still there are: the conflict between the two parameter update methods, the missing details about the RL task itself, questions about whether the experiments are fair and can be reproduced, and the overall writing quality making it tough to follow.

**Reviewer Scores:**

Without a rebuttal, the scores probably wouldn't move.

---

### Decision · Program_Chairs · 2026-01-26

Reject